

# The multi-year negative Indian Ocean Dipole of 2021-2022

Ankur Srivastava[1, *], Gill M. Martin[2], Maheswar Pradhan[1], Suryachandra A. Rao[1] and Sarah Ineson[2]

[1] Indian Institute of Tropical Meteorology, Ministry of Earth Sciences, Pune, India
[2] Met Office, Exeter, UK

*Correspondence to*: Ankur Srivastava (ankur@tropmet.res.in)

**Abstract.** The years 2021 and 2022 witnessed negative Indian Ocean Dipole (nIOD) conditions, with the 2022 event being the strongest on record. The dipole mode index was negative since the summer of 2021 and remained negative until early winter 2022, an unprecedented duration of 19 months. This makes it the first such occurrence of a multi-year nIOD. It co-existed with a triple-dip La Niña event during 2020-2022. In this study, we explore the dynamics behind the occurrence of

this multi-year nIOD event. The tropical Indian Ocean (TIO) witnessed predominant westerly wind anomalies starting in the summer of 2021 and lasting till the end of 2022, with a record number and duration of westerly wind bursts (WWBs). The anomalous westerlies were supported by the background La Niña state and anomalous convection over the eastern TIO associated with tropical intra-seasonal oscillations. Occurrences of WWBs outside their preferred climatological months and strong westerly wind anomalies modulated the intensity of the zonal currents and the Wyrtki jets in the TIO. The associated

heat and mass transfer caused the depression of the thermocline in the eastern TIO, resulting in the sustenance of nIOD conditions. Anomalous westerly wind activity in the TIO during the spring of 2022 served as a bridge between the two nIOD events and sustained it for a record duration. This multi-year nIOD event thus prevented the Indian summer monsoon rainfall from being in large excess, as the monsoon conducive modulation of the Walker circulation was counteracted by the anomalous subsidence over India by the nIOD-modulated regional Hadley circulation.

## 1 Introduction

The Indian Ocean Dipole (IOD) is a strongly coupled ocean-atmosphere phenomenon that manifests as a zonal dipole in the Indian Ocean sea-surface temperature (SST) (Ajayamohan et al., 2009; Ajayamohan & Rao, 2008; Rao et al., 2009; Rao & Yamagata, 2004; Saji et al., 1999). Anomalously warm (cool) SST in the eastern (western) Indian Ocean is a signature of a negative IOD (nIOD) event. nIOD events tend to suppress the Indian Summer Monsoon rainfall (ISMR; Ajayamohan et al.,

2008, 2009; Ajayamohan & Rao, 2008; Ashok et al., 2001; Cherchi et al., 2007; Cherchi & Navarra, 2013; Krishnan et al., 2011; Ratna et al., 2021), can cause severe droughts over East Africa during the short rainy season (Black et al., 2003; Endris et al., 2019; Manatsa & Behera, 2014), Indonesia (Nur'utami & Hidayat, 2016), Australia (Cai et al., 2009), impacts the Sri Lankan rainy season during September to December (Zubair et al., 2003), and exerts remote influences over South China and Brazil (Bazo et al., 2013; Chan et al., 2008; Qiu et al., 2014; Taschetto & Ambrizzi, 2012). Therefore, the evolution of

IOD is closely monitored by operational weather and climate forecasting centers.



IOD conditions in the Indian Ocean are quantified using the dipole mode index (DMI), which is defined as the difference in area-averaged SST anomalies over western (50°E-70°E, 10°S-10°N) and eastern Indian Ocean (90°E-110°E, 10°S-Equator), standardized by its standard deviation. Strong nIOD conditions evolved during the boreal summer of 2022. DMI became strongly negative with a value less than -2 standard deviations during September-November (SON) 2022, thereby making it

one of the strongest nIOD events on record (Fig. 1a). It is also worthwhile to note that nIOD-type conditions were present during the summer of 2021 as well, with the DMI index (3-month running mean, Fig. 1b) being negative from May 2021 and lasting till the end of 2022. Such persistent nIOD conditions, which lasted for about 19 months, have never been associated with other nIOD events in the observational record since 1960 (since when reliable Indian Ocean observations became available, Fig. 1d). Another peculiar feature of this event was that it peaked during the boreal summer of 2022 (Fig. 1b),

unlike other events, which usually peak in SON. The 2021-22 event co-occurred with a triple-dip La Niña event, which developed in mid-2020 and continued till 2022 (as is seen from the Niño 3.4 index, Fig. 1c). Such triple-dip events are rare and have occurred only four times since 1950 (Ratna et al., 2024). Multi-year La-Niña events exhibit an increasing trend, with eight out of the ten events in the past century have occurred after 1970 (Wang et al., 2023).







**Figure 1: (a) The September-November (SON) mean DMI index, and the IOD east and west poles, for the period 1960-2022 based on ERSST dataset; (b) The monthly DMI index from ERSST (bars), and the three-month running mean for ERSST, OISST and HadlSST (lines) for the period 2020-2022; (c) same as (b) but for the Nino 3.4 index; and (d) The time evolution of the 3-month running mean of DMI based on ERSST dataset for all nIOD events since 1960. (-1) on the x-axis indicates the year previous to the nIOD event, and (+1) indicates the subsequent year. The black curve denotes the evolution of DMI from NDJ 2021 to OND 2023.**

Strong La Niña conditions often result in a large excess in rainfall over India during the summer monsoon season (often exceeding 110% of the long-period average (LPA) during June-September (JJAS) due to the modulation of the associated Walker circulation (Rasmusson et al., 1983; Sikka & Gadgil, 1980). The persistent La Niña conditions during 2020-22, however, did not result in a very large positive rainfall anomaly over India. 2020 was an above-normal monsoon year (109% of the long-period average, LPA). 2021 (99% of LPA) was a normal monsoon year, and rainfall during 2022 was above normal (106% of LPA; Ratna et al., (2024)). The lack of a large positive rainfall anomaly in these years might have been caused by the compensating effect of the nIOD conditions, as anomalously warm SST in the IOD east pole can suppress the convection over India by the modulation of the local Hadley circulation.

The interaction between the El Niño and Southern Oscillation (ENSO) and IOD is well-documented. Model experiments have revealed that about one-third of the IOD events can be explained by ENSO forcing, while the rest of the events develop due to the internal dynamics of the Indian Ocean (Behera et al., 2006; Xiao et al., 2022; Yang et al., 2015). An ENSO event can also be forced by an extreme IOD event (Annamalai et al., 2005; Luo et al., 2010). For the 2021-22 nIOD case, the evolution of La Niña conditions in the Pacific preceded the development of nIOD. This leads to the assumption that the nIOD event might be forced by tropical Pacific SSTs. Negative (positive) IOD events tend to co-occur with La Niña (El-Niño) as the associated modulation of the Walker circulation forces anomalous westerly winds (easterlies) in the tropical Indian Ocean (TIO) region. These anomalous winds are reinforced by local air-sea interactions, which peak in the boreal autumn (Hendon, 2003; Hendon et al., 2012). Anomalously strong equatorial westerly winds, often called westerly-wind bursts (WWBs), can trigger strong eastward current in the equatorial Indian Ocean, referred to as the Wyrtki jets (WJs) (Wyrtki, 1973). These jets distribute upper-ocean heat and mass (Reverdin, 1987; Wyrtki, 1973), and anomalous changes to the jets are instrumental in the development of IOD events (Cai et al., 2014; Murtugudde et al., 2000; Vinayachandran et al., 1999, 2007). Was the triple-dip La Niña of 2020-2022 responsible for maintaining nIOD conditions for a record 19 months?

The 2021-22 nIOD event was, therefore, unique in several aspects: (1) It was the strongest nIOD event recorded to date. (2) It peaked during the boreal summer of 2022. (3) nIOD conditions were sustained for a record 19 months, making it the first occurrence of a multi-year nIOD event. (4) This event co-occurred with a triple-dip La Niña event.

These peculiar aspects of the event warrant further investigation of the dynamics leading to such a long-lived event. This assumes further significance as the prolonged duration of extreme climatic events can pose hazards comparable to the





intensity of these events. Recent studies have shown that the frequency of consecutive La Niña events has increased in recent years (Wang et al., 2023) and is projected to increase further in a warming world (Geng et al., 2023). Consecutive La Niñas can intensify nIOD conditions in the TIO, therefore compounding the climate risk associated with such events. In this study, we aim to investigate the dynamics behind the multi-year nIOD event of 2021-22. The possible impact of the triple-dip La Niña of 2020-2022 and the multi-year nIOD on the Indian monsoon is also investigated.

The article is organized as follows: Section 2 describes the observational data used for the analysis and the methodology. Section 3 describes the major findings, and Section 4 summarizes the results.

## 2 Data and Methods

The following sources of observational monthly SST are utilized: (1) the Extended Reconstructed Sea Surface Temperature (ERSST), Version 5 (Huang et al., 2017), (2) the Optimum Interpolation Sea Surface Temperature (OISST) version 2.1

(Huang et al., 2020), and (3) the Hadley Centre Sea Ice and Sea Surface Temperature data set (HadlSST; Rayner et al., 2003). The ocean temperature profiles and currents (pentad) are obtained from the National Centers for Environmental Prediction (NCEP) Global Ocean Data Assimilation System (GODAS; Behringer et al., 1998), and the depth of the 20 °C isotherm is computed from it. The monthly temperature profiles from the MOAA GPV 1-degree gridded dataset (Hosoda et al., 2008) produced by the Japan Agency for Marine-Earth Science and Technology (JAMSTEC) are also used to compute

the 20 °C isotherm. The daily zonal and meridional winds are obtained from the fifth-generation European Centre for Medium-Range Weather Forecasts (ECMWF) reanalysis (ERA5; Copernicus Climate Change Service, 2017, DOI:10.24381/cds.adbb2d47). The daily Southern Oscillation Index (SOI) data for the period 1991-2022 (calculated using the 1887-1989 base period) is obtained from the open data portal of the Queensland Government (from their website at https://data.longpaddock.qld.gov.au/SeasonalClimateOutlook/SouthernOscillationIndex/SOIDataFiles/LatestSOI1887-

1989Base.txt; last accessed 03/01/2024). The observational phases and amplitudes of the Madden-Julian Oscillation (MJO) and Boreal Summer Intraseasonal Oscillation (BSISO) are based on Kikuchi (2020) and Kikuchi et al., (2012), and are obtained from the website of the International Pacific Research Center (https://iprc.soest.hawaii.edu/users/kazuyosh/ISO_index/data/MJO_25-90bpfil.rt_pc.txt; https://iprc.soest.hawaii.edu/users/kazuyosh/ISO_index/data/BSISO_25-90bpfil.rt_pc.txt; last accessed 03/01/2024). The

daily Altimeter satellite gridded Sea Level Anomaly (SLA) level 4 data is obtained from the Copernicus Marine Service website (https://doi.org/10.48670/moi-00148).

The WWB events in the Indian Ocean are identified from ERA5 10 m zonal winds following the methodology described in Seiki and Takayabu (2007), but with a more stringent check on the duration of WWBs. Daily anomalies of the 10m zonal

winds averaged between 2.5°S-2.5°N were computed as the deviations from a 91-day running mean daily climatology. The cases where the anomalies met or exceeded a threshold of 5 m s$^{-1}$ with a longitudinal extent of 10° longitude (~1100 km) and





lasted for more than 4 days (as compared to 2 days in Seiki and Takayabu (2007)) were classified as WWBs. The time duration between two consecutive WWBs should be more than 7 days to ensure that the two events are independent of each other and not a part of a common westerly wind activity.


## 3 Results

### 3.1 Role of Winds, Westerly Wind Bursts, and Wyrtki Jets

Negative IOD events are often triggered by equatorial westerly wind anomalies in the TIO (Ashok et al., 2004; Saji & Yamagata, 2003). Fig. 2a shows the 11-day running mean of equatorial (averaged over 5°S-5°N, 60°E-90°E) 10m zonal-

wind anomalies for all the nIOD events since 1980. As the 2022 event is thought to be a multi-year event, the wind anomalies for the previous two years (2020 and 2021) are also shown in the same figure (indicated by year-1 and year-2 in Fig. 2a).



**Figure 2: (a) The 11-day running mean of equatorial (averaged over 5°S-5°N, 60°E-90°E) 10m zonal-wind anomalies for all the nIOD events since 1980. The wind anomalies for the previous two years are indicated by Year -2 and Year -1 on the x-axis. (b) Evolution of the equatorial 10m zonal-winds wind anomalies (dashed line), its 7-day running mean (solid line) for the period January 2020 to December 2022.**

Each nIOD event has a unique evolution of zonal winds, with strong westerly wind anomalies generally favored from mid-summer (July-August), persisting till the end of the year. A spell of westerly wind anomaly starting in October 2020 and lasting till January 2021 is particularly interesting. Climatologically, the zonal surface (850 hPa) winds over EIO during January-February are close to zero (weak easterly). However, January 2021 witnessed strong anomalous westerly winds, with peak anomalies of ~5 m s⁻¹. There were multiple strong and long-lived westerly wind events during 2021-2022. The





wind anomalies were almost always westerly, starting in mid-May 2021 till December 2022, with a few weak and short-lived anomalous easterly events (Fig. 2b). None of the other nIOD events have exhibited such a strong and persistent westerly wind forcing. The zonal winds at 850 hPa also exhibit similar signatures (figure not shown).

WWBs are short-duration synoptic-scale disturbances that occur near the equator and are found over the Indian and Pacific Oceans. Past studies suggested that WWB occurrences are random (Fedorov et al., 2003; Moore & Kleeman, 1999); however, more recent studies have shown that they are often associated with large-scale environmental conditions such as ENSO (Eisenman et al., 2005; Lengaigne et al., 2003, 2004; Roundy & Kiladis, 2006; Seiki & Takayabu, 2007). WWBs are defined following Seiki and Takayabu (2007; see Section 2 Data and Methods), and the annual count and cumulative duration of WWBs for 1980-2022 are shown in Fig. 3a. 2021 witnessed a record number of five WWBs with a cumulative duration of ~29 days, while 2022 witnessed three events with a cumulative duration of ~23 days (third largest duration since the 1980s; Fig. 3a).

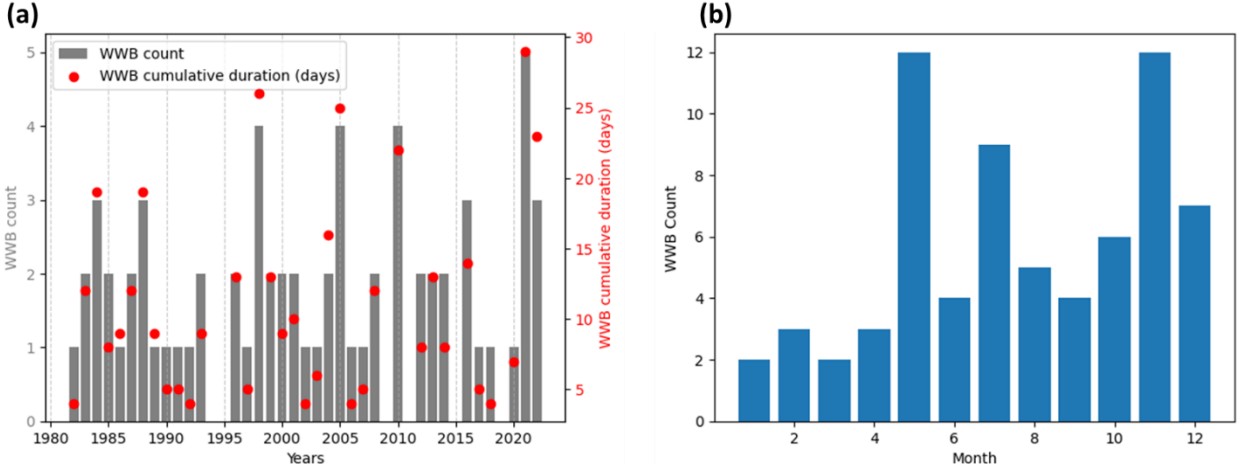

**Figure 3: (a) Number (bars) and cumulative duration (days, red markers) of WWBs; (b) Total WWB count by month for the years 1980-2022 to illustrate the seasonal cycle.**

Boreal spring and fall are the transition seasons between the southwest and northeast monsoons. Strong westerly winds during these seasons drive strong eastward flow in the upper 80-100m of the equatorial Indian Ocean and are referred to as the Wyrtki jets. They play an important role in heat and mass transfer in the Indian Ocean. McPhaden et al. (2015) demonstrated the role of WJs in IOD development. Recently, Xiao et al. (2024) have reported an association between easterly wind bursts and positive IOD events. Given the record number and duration of WWBs in the Indian Ocean during 2021-2022, the peculiar features of WJs are explored in the subsequent sections.



Fig. 4 shows the time-longitude sections of the upper-ocean state, associated surface winds, and the WWB occurrences. The evolution of La Niña, WWBs, Wyrtki jets, depth of the 20 °C isotherm (D20), and BSISO/MJO is shown in Fig. 5. The

160   occurrences of WWBs (dotted vertical blue lines) and the event numbers (in blue circles, top panel) are also indicated in Figs. 4 and 5.  The succeeding discussion is arranged around the event numbers as shown in Figs. 4 and 5.

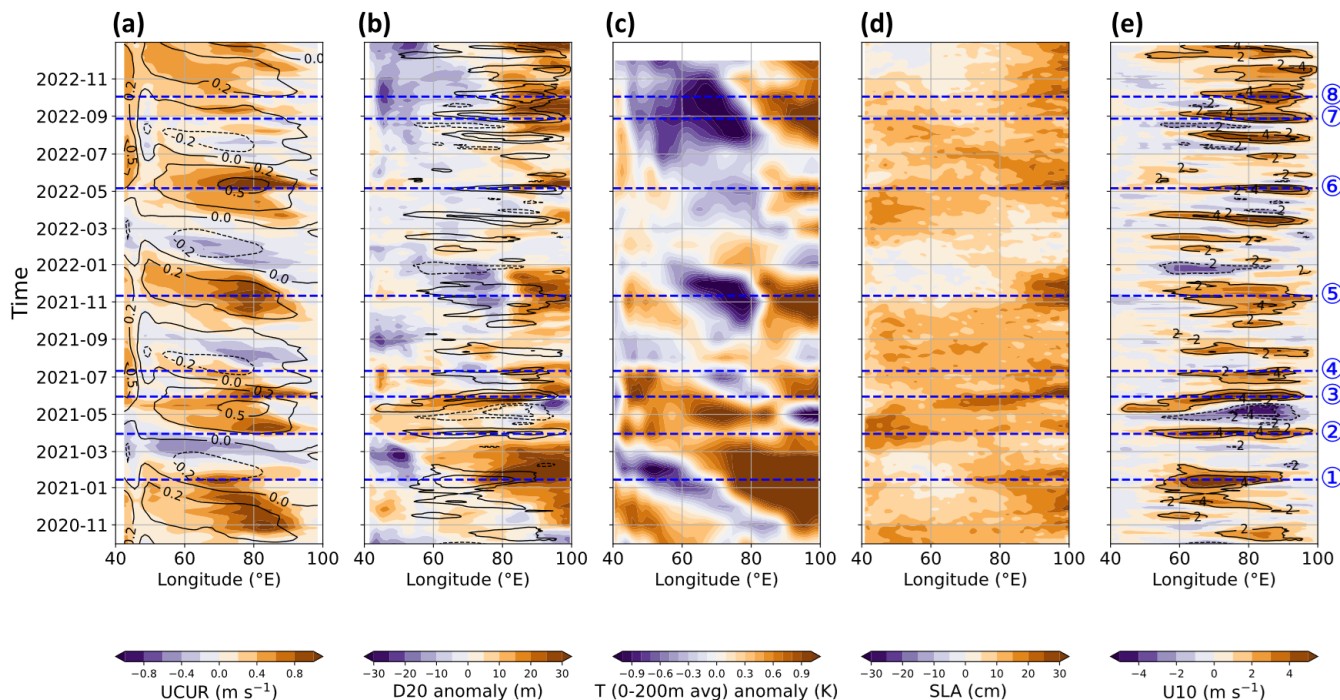

**Figure 4: The time-longitude sections of (a) 2°S-2°N averaged zonal currents in the upper ocean (0-100m averaged)**

165   **for 40°E-100°E along with the climatological currents (contours), (b) The D20 anomaly (m) from NCEP GODAS (shading), and the 11-day running mean of 10 m zonal wind anomaly (m s⁻¹) averaged over 5°S-5°N (positive (negative) values are indicated by solid (dashed) contours and the contour interval is -4 to 4 by 2, zero contour is not shown), (c) the 0-200m averaged weekly temperature anomaly from NCEP GODAS, (d) the observed sea-level anomaly (SLA) from AVISO, and (e) 11-day running mean of zonal wind (shading) and zonal wind anomaly**

170   **(contours).**



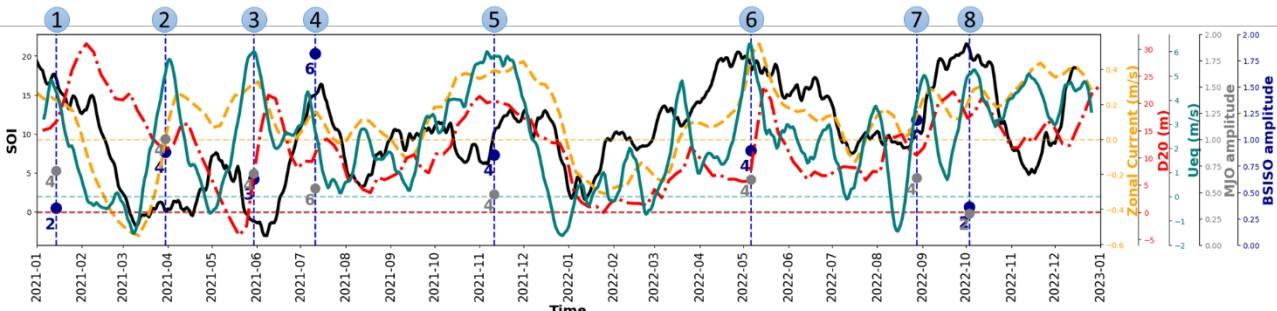

**Figure 5: The time evolution of 31-day running mean of daily SOI, equatorial zonal currents (OSCAR, 2°S-2°N, 40°E-100°E), GODAS D20 anomalies over the IOD east pole, and the 10 m zonal wind anomalies (5°S-5°N, 60°E-90°E). The vertical dotted lines in blue indicate the occurrences of WWBs, and the MJO (gray)/BSISO (blue) phases on the day of WWBs are indicated by markers.**

1. Climatologically, the spring WJ is restricted to April-May, while the fall WJ lasts longer from mid-September to early-January. As mentioned earlier, January 2021 witnessed strong westerly wind activity in the equatorial TIO (Fig. 2, 5). In response to this unusual occurrence of a WWB in January (marked as event 1 in Fig. 4, which has occurred only twice since 1980, Fig. 3b), the fall WJ strengthened in early January instead of decaying (Fig. 4a, 5). This unusual forcing was, in fact, so strong that it led to a positive D20 anomaly exceeding 40m in the IOD east pole region (Fig. 4b, c, 5), a build-up of significant heat content in the upper ocean (Fig. 4c), and positive sea-level anomaly (SLA, Fig. 4d). This positive anomaly in the upper-ocean heat content lasted till the end of March 2021. This unusual occurrence of the WJ in January 2021 was due to the formation of a low-pressure system off the coast of Sri Lanka. This system led to widespread extreme rainfall over parts of Southern India. The northeast monsoon season usually ends by December, but the unusual January rain extended the season till mid-January (India Meteorological Department, 2021).

2. The spring of 2021 saw two WWBs separated by two months. The first occurred towards the end of March (event 2), and the other was close to monsoon onset over Kerala in June (event 3), initiating WJs (Fig. 4a, 5). In response to event 2, the anomalous D20 deepened slightly before witnessing a rapid shoaling towards the end of May 2021 (Fig. 4c, 5). The WJs usually peak around the Bay of Bengal monsoon onset date in early May (Li et al., 2022) and gradually decelerate thereafter.

3. Event 3 led to the extension of the WJ outside the usual period of April-May (Seiki and Takayabu, 2007; McPhaden et al., 2015), lasting till mid-June. It caused an eastward propagating, downwelling Kelvin wave, which caused rapid deepening of D20 (reaching a positive anomaly of ~20m) and an increase in SLA over the eastern TIO due to the strengthening of the WJ (Fig. 4b, c, d, 5). The transition of equatorial westerlies to cross-equatorial monsoon flow causes the WJ to dissipate. The progression of monsoon after the onset over Kerala on 03[rd] June 2021 was sluggish due



to the co-occurrence of cyclone Yaas on May 26 (close to the monsoon onset), and a delay in transition to cross-equatorial monsoon flow. This delay might have provided conducive conditions for the WJ to last till mid-June.

4.   July 2021 witnessed another WWB (event 4), thus causing the zonal surface currents to be weakly eastward (~0.2-0.4 m s$^{-1}$) and a positive SLA. This WWB maintained the D20 anomaly around ~14 m. During mid-August, another bout of anomalous zonal westerly winds was observed (with a magnitude of ~2 m s$^{-1}$). This event was not strong enough to qualify as a WWB but sustained the weak eastward current. The sequence of persistent zonal westerly wind anomalies starting in mid-May 2021 till the end of September played an important role in modifying the seasonality of the upper-

ocean zonal currents such that they remained weakly eastward till late July, against the climatological transition to westward in mid-June. The associated mass transport caused the thermocline to uplift in the western TIO and deepen in the eastern TIO. The associated weakening of climatological south-easterlies along the Java-Sumatra coast during boreal summer reduced the evaporative cooling and maintained the positive SST anomaly. Thus, the negative IOD conditions were maintained during the boreal summer of 2021.

5.   Starting in mid-September, an equatorial westerly wind anomaly was observed, which lasted till early December (Fig. 4b, e). The fall WJ initiated towards the end of September and peaked in mid-November (Fig. 4a, event 5). This led to the deepening of D20 in the eastern TIO (Fig. 4 b, c, 5), positive SLA (Fig. 4d), and nIOD conditions persisted till the end of 2021. The larger intra-seasonal variability in the anomalous westerly wind activity in boreal spring compared to boreal fall is worth noting. The boreal fall WJs usually subside towards the end of January next year, and weak

westward flowing surface currents (with a climatological magnitude of ~0.2 m s$^{-1}$) are observed in the equatorial TIO during February-March.

        The anomalous zonal westerly winds subsided in early December 2021 (Fig. 2, 4e, 5), in response to which the fall WJ also weakened towards the end of December 2021 (Fig. 4). Thus, the positive D20 anomaly also started decreasing and remained close to zero during January-February 2022 (Fig. 4 b, c, 5). Two peaks in anomalous zonal westerly winds

occurred during February-March 2022, with peak magnitudes in the range of 3-4 m s$^{-1}$ (Fig. 2, 4b, 5). These were not strong enough to qualify as WWB since the zonal extent was not basin-wide but restricted to the eastern TIO. However, they possibly weakened the westward-flowing zonal ocean current. The second peak initiated the eastward ocean current in mid-March 2022 and caused the deepening of the thermocline in the eastern TIO (by ~10m towards the end of March, Figs. 4 b, c, 5).

6.   The current was strengthened by a WWB event occurring in early May and triggered a WJ (event 6). Intermittent bouts of anomalous zonal westerly winds in the TIO from May to December 2022 were observed, with peaks occurring in early May, early June, early July, early August, late August, early October, mid-November, and December (Fig. 5). Climatologically, during the summer monsoon months of JJA, the zonal current in the upper TIO is weak westward, and the current reverses from westward to eastward in mid-September (Fig. 4a). In response to the westerly zonal wind

anomaly, which was initiated in late July and peaked in early August, the sign of the current reversed in early August (Fig. 5).

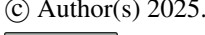



7.  The two subsequent WWBs (late August and early October, events 7 & 8) initiated the fall WJ in early September, with peak surface currents in November 2022. Eastward zonal currents lasted about five months (August-December 2022) with large intra-seasonal fluctuations, thus maintaining the nIOD conditions.


The interplay between the persistent zonal westerly wind anomalies, WWBs, the WJs, and the associated deepening of the thermocline sustained the nIOD conditions starting in May 2021 and lasting till November 2022 – a record 19 months! The 2022 nIOD was stronger than the 2021 event, possibly because of three reasons – (1) the 2021 event pre-conditioned the TIO, (2) strong zonal westerly wind anomalies were observed during February-March 2022 with the associated weakening

of the westward zonal current, and (3) The large intra-seasonal variability of zonal westerlies during summer and fall of 2022 which had a larger amplitude compared to those during 2021.

### 3.2 Impact of La Niña, BSISO, and MJO on the 2021-2022 nIOD

It is clear from the preceding discussion that the westerly wind forcing in the TIO played an important role in the sustenance of nIOD conditions during 2021-22. It has been shown in previous studies that ENSO and IOD events have a strong

tendency to co-occur, though they can evolve independently as well. Seiki and Takayabu (2007) noted that WWBs in the Indian Ocean occur in phase with La Niña events and rarely occur during El-Niño years. The prolonged La Niña conditions during 2020-2022 could have provided a conducive background state to the frequent WWB activity in the TIO. To explore this aspect, a scatter plot of the 31-day running mean of the Southern Oscillation Index (SOI) and the zonal wind speed averaged over 2.5°S-2.5°N, 60°E-90°E on the day of peak WWB activity is shown in Fig. 6. The 31-day running mean of

daily SOI for the period 1991-2022 is considered for this purpose. The individual nIOD events (1996, 1998, 2005, 2010, 2016) and the 2021-22 event are indicated. Also shown is the linear regression line. Sustained SOI value greater (lesser) than +7 (-7) indicates La Niña (El-Niño) conditions (Yu et al., 2021). Daily values of SOI reflect the day-to-day fluctuations of weather patterns but are not representative of the background ENSO state. Thus, we use a 31-day running mean of daily SOI values, such that the daily weather fluctuations are filtered out, but some elements of the intra-seasonal fluctuations of

the Walker circulation are retained. Strong La Niña conditions modulate the zonal Walker circulation such that there is anomalous subsidence over the central and eastern Pacific while there is anomalous convection over the Maritime continents. This anomalous convection can strengthen the zonal westerly wind anomalies in the TIO. It is clear from Fig. 6 that not all WWB/westerlies are associated with La Niña. Many WWB events lie within the ±7 envelope, which might not be associated with La Niña forcing. However, a sizeable number of WWB events occur when SOI is greater than +7, and

almost no WWB events occur when SOI is less than -7, which is associated with the El-Niño forcing. Five out of the eight WWBs during 2021-22 occurred when the SOI was greater than +7. From Fig. 5, it is seen that some of the peaks of SOI match well with the peaks of westerly wind activity, for example, the peak during July 2021, May 2022, and October 2022. In other instances, the WWB activity occurred during the strengthening/mature phase of the SOI. Thus, we infer that the background state associated with the La Niña provided conducive conditions for triggering WWBs in the TIO.






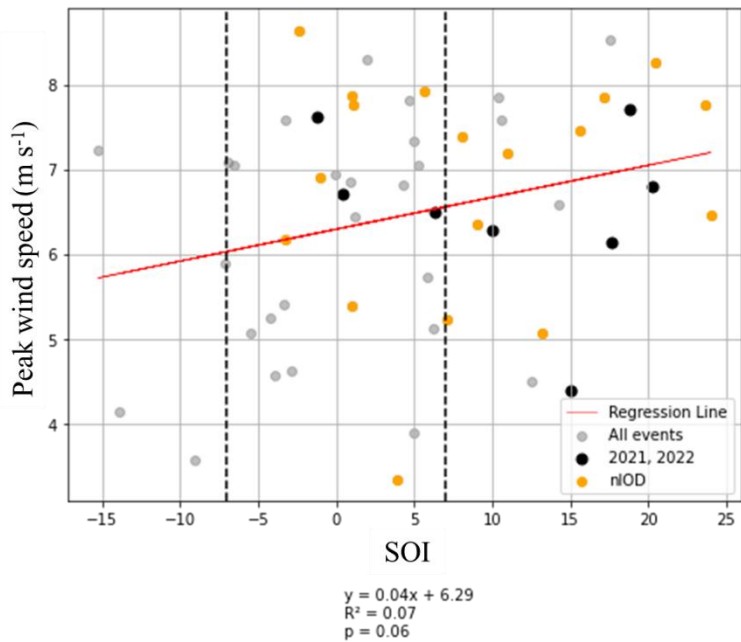

**Figure 6: Scatter plot of the 31-day running mean of the daily Southern Oscillation Index (SOI) for the period 1991-2022 with the peak zonal winds averaged over 2.5°S-2.5°N, 60°E-90°E associated with WWBs. The regression equation is indicated.**


The large intra-seasonal variability in tropical convection can be attributed to the equatorial eastward propagating Madden-Julian Oscillation (MJO) and the northward propagating boreal summer intra-seasonal oscillation (BSISO) in the off-equatorial monsoon regimes. The interaction between IOD and BSISO/MJO is well known (Rao et al., 2009; Shinoda and Han, 2005). Out of the eight WWBs during 2021-22, six out of eight events were associated with phase 4 of MJO (Fig. 5),

when the convection is located over the eastern Indian Ocean. Three out of the four events occurring during boreal summer co-occurred with phases 3 or 4 of the BSISO (Fig. 5), during which convection is centered over the eastern TIO and western Pacific, respectively, and are associated with enhanced westerlies in the equatorial IO. Interestingly, these BSISO phases were supported by a peak (three events)/rapidly increasing values of SOI (two events), indicating a possible association between the ENSO state and the BSISO.

**3.3 Anomalous sub-surface conditions**

Fig. 7 shows the monthly time series of standardized D20 anomalies averaged over the IOD east pole region from MOAA GPV data (2001-2022) and NCEP GODAS (1980-2022). A positive D20 anomaly indicates downwelling. Positive D20 anomalies dominate during 2021-22 in both datasets. In fact, such a persistent downwelling episode has not been observed



since the 1980s. The scatter plots of standardized D20 anomalies over the IOD east pole and the 10m zonal equatorial wind

anomaly for each season are shown in Fig. 8. The individual nIOD events and pIOD events are also indicated. The

standardized D20 anomaly remained greater than +0.5 (often greater than or equal to +1) for all the seasons during 2021 and

2022, except for December-February (DJF) 2022. The associated anomalous zonal wind forcing was also very strong (often

greater than +1 standard deviation). The positive D20 and surface zonal wind anomalies during DJF and March-May (MAM)

seasons during 2021 and 2022 are unlike other nIOD events, where such positive anomalies are seen only during June-

August (JJA) and SON. The persistent westerly wind anomalies and the WJs prevailed and conditioned the Indian Ocean to

an nIOD state by suppressing the upwelling along the Java-Sumatra coast (Schott et al., 2009).

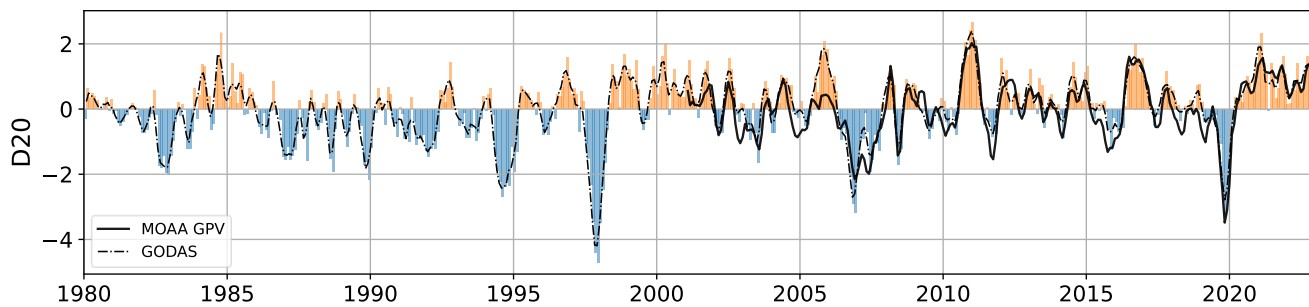

**Figure 7: The evolution of the 3-month running mean of the standardized anomalies of the depth of the 20°C**

**isotherm (D20) over the IOD east pole region from NCEP GODAS (dashed lines; 1980-2022), and MOAA GPV data**

**(solid lines; 2001-2022). The bars indicate the monthly anomalies for the NCEP GODAS dataset.**





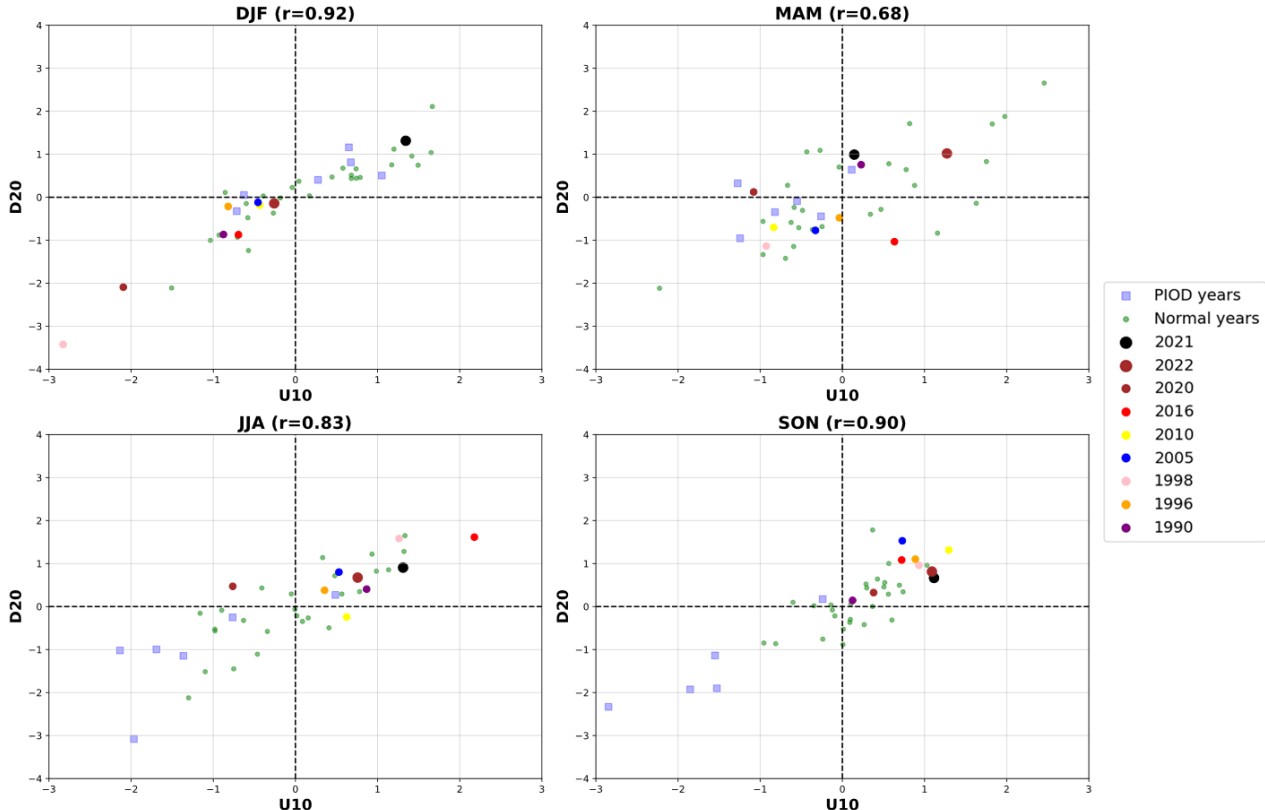

**Figure 8: Scatter plot of the standardized anomalies of 10 m zonal winds over the equator (U10; averaged over 5°S-**
**5°N, 60°E-90°E) and D20 anomalies over the IOD east pole region for different seasons. The correlation (r) between**
**the time series is indicated in brackets.**

The absence of climatological upwelling in the eastern equatorial TIO is due to the lack of steady climatological easterlies.
The climatological south-easterlies from April to October off the Java-Sumatra coast favour coastal upwelling and can
trigger a pIOD event. During nIOD events, this upwelling is suppressed. Positive D20 anomalies indicate the suppression of
upwelling in the IOD east pole region starting in late 2020 and lasting till the end of 2022 (Figs. 7 and 8). The coupling
between SST and D20 in the IOD east pole is positive only during the boreal summer (r=0.59) and fall seasons (r=0.68), as
indicated by the scatter plots of SST and D20 in Fig. 9. The large sub-surface heat content during DJF and MAM 2021 was
therefore not reflected in the SST. During JJA 2021, when the SST-D20 coupling is positive, positive SST anomalies
emerged in the IOD east pole and were maintained during SON 2021. Positive D20 anomalies (greater than +1 standard
deviation) again emerged during MAM 2022 with weakly positive SST anomalies. These positive SST anomalies peaked in
JJA 2022, coinciding with the DMI peak, and diminished during SON 2022.





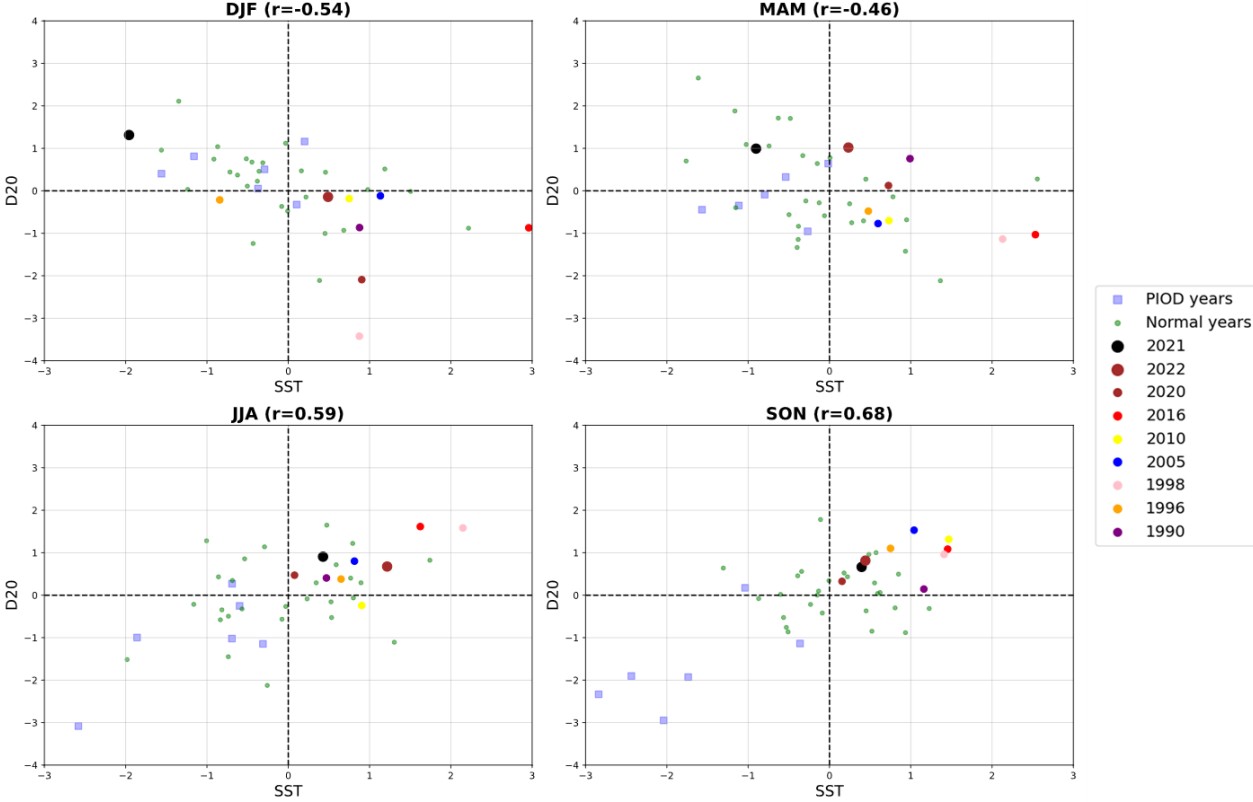

**Figure 9: Scatter plot of the standardized anomalies of SST and D20 anomalies over the IOD east pole region for different seasons. The correlation (r) between the time series is indicated in brackets.**

**3.4 Impact on Indian Summer Monsoon**

The June-September (JJAS) mean Walker and Hadley circulation derived from ERA5 for 2021 and 2022 is shown in Fig. 10. The La Niña conditions during JJAS 2021 were weak, with the monthly Niño 3.4 index remaining less than -0.5 standard deviation. Therefore, the associated ascending branch of the Walker cell was restricted to the warm pool region longitudes (~120°E-180°E, Fig. 10a). Weak nIOD conditions in the TIO forced anomalous ascending motion over the IOD east pole region in the regional Hadley circulation and caused subsidence over northern Indian latitudes (Fig. 10c). The combined influence of weak La Niña and nIOD during JJAS 2021 prevented above normal monsoon rainfall over India (JJAS rainfall being ~99% of LPA). Strong La Niña conditions were present during the summer of 2022. It caused large-scale modulation of the Walker circulation such that strong ascending motion is noted over Indian longitudes (60°E-90°E, Fig. 10c). Strong nIOD conditions modulated the regional Hadley circulations and caused ascending motion over the IOD east pole with subsidence over Indian latitudes. Thus, nIOD conditions suppressed the influence of strong La Niña present during JJAS 2022, and the resultant rainfall over India during JJAS was restricted to ~106% of LPA.



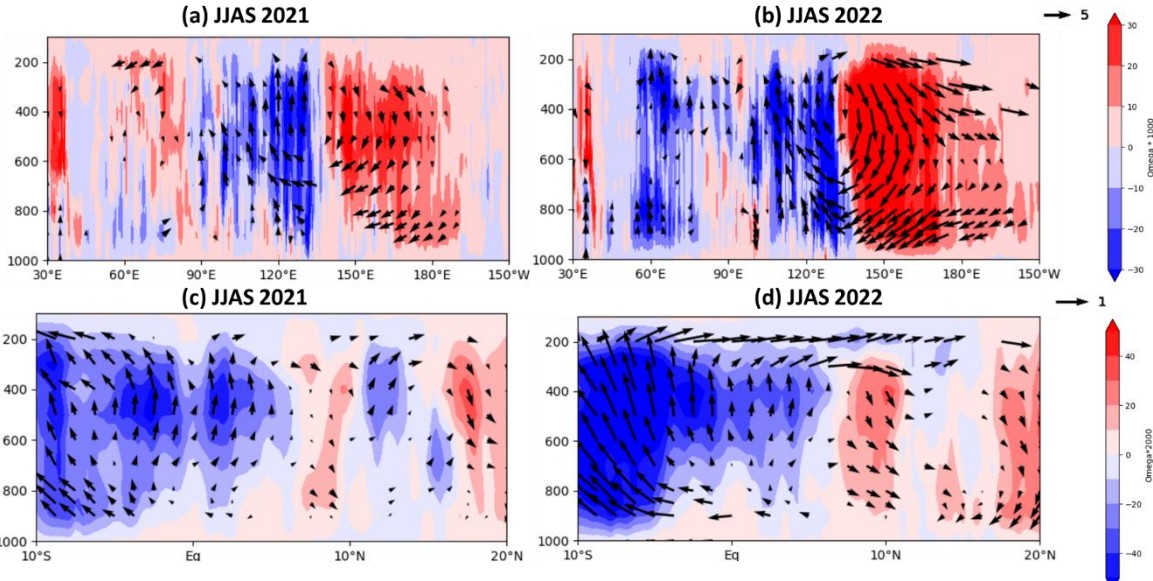

**Figure 10: The height-longitude section of the anomalous JJAS mean Walker circulation (averaged over 1°S-1°N) for (a) 2021 and (b) 2022, and the height-latitude section of the anomalous JJAS mean Hadley circulation (averaged over 90°E-110°E) for (c) 2021 and (d) 2022. The vertical velocity (Omega) is scaled by 1000 and 2000 in (a, b) and (c, d), respectively, to elucidate the vertical motion. Vector magnitudes less than 10 and 5 are masked out in a, b, and c, d, respectively.**

## 4 Discussion and Summary

The nIOD event of 2022 was one of the strongest events in the observational record. The event was unique in several aspects. The DMI was negative since May 2021 and remained negative until the end of 2022, making it the first ever multi-year nIOD event on record since the 1960s. It co-occurred with the triple-dip La Niña of 2020-2022. 2021-2022 witnessed a record number of WWBs in the TIO, with the largest cumulative duration. Despite a persistent La Niña forcing in the tropical Pacific, the Indian Summer Monsoon rainfall during 2021-22 did not witness a large excess. Did the multi-year nIOD event have a compensating impact on the ISMR? Motivated by this question, we explore the dynamics behind the first-ever multi-year nIOD event ever recorded. The major findings are summarized below:

1. 2021-22 witnessed large intra-seasonal fluctuations in the 10 m equatorial zonal wind anomalies, with the anomalies mainly remaining westerly from May 2021 to November 2022. Such persistent zonal westerly wind anomalies are not seen to be associated with other nIOD events.





2.  2021 witnessed the largest WWB activity in the TIO. Most unusual was the WWB occurrence in January 2021, which led to a build-up of a large positive heat content anomaly in the eastern TIO. The WWBs during the boreal spring of 2021 initiated the positive D20 anomaly over the eastern IO. The WWB during July 2021 and the subsequent anomalous westerly winds maintained a weak eastward equatorial zonal current during the summer of 2021. This maintained the positive D20 anomaly in the eastern IO, which strengthened during the fall of 2021.

3.  Strong and unusual anomalous westerly winds over TIO during February-March 2022 weakened the climatological westward current and initiated a spring WJ during mid-March 2022. This jet peaked in May 2022 in response to a WWB, and the eastward zonal current lasted till June. These sequences of events deepened D20 in the eastern TIO during MAM 2022. Two large-amplitude anomalous westerly wind events during August and September initiated weak eastward zonal currents, which were strengthened in the fall on account of the WJ dynamics. Thus, the nIOD event was maintained till the end of the year.

4.  WWBs played an important role in sustaining the multi-year nIOD event. La Niña forcing played an important role in maintaining conducive conditions for WWBs. Out of the eight WWB events during 2021-22, five occurred when SOI was greater than +10, suggesting the role of La Niña forcing. Some of the peaks in SOI match well with the peaks of anomalous westerly wind activity, while in other instances, the WWB activity occurred during the strengthening/mature phase of the SOI.

5.  Six out of the eight WWB events during 2021-22 co-occurred with phase 4 of the MJO, and three out of the four events occurring during boreal summer co-occurred with phase 3 or 4 of the BSISO. These MJO/BSISO phases aid zonal westerly wind anomalies in the TIO, as the convection is located over the eastern Indian Ocean.

6.  The co-existence of La Niña and nIOD conditions had a compensating impact on the Indian monsoon. Weak La-Niña and nIOD conditions caused a near-normal monsoon rainfall during 2021. During 2022, when La Niña was much stronger, the strong nIOD conditions induced anomalous subsidence over the Indian landmass by modulating the regional Hadley circulation. This prevented the ISMR from being in large excess.

To conclude, the sustenance of the multi-year nIOD event was the outcome of various factors that co-existed and complemented each other. Most importantly, the triple-dip La Niña provided a conducive background state in the Indian Ocean for nIOD to develop. The La Niña modulated Walker circulation, resulting in frequent WWBs in the TIO. WWBs and equatorial zonal westerly wind anomalies were also aided by the MJO/BSISO. Anomalous convection in the eastern TIO during February-March 2022 caused strong anomalous zonal westerly winds in the TIO, leading to the early initiation of Wyrtki jets and not allowing the positive D20 anomaly in the eastern TIO to subside. It, therefore, acted as a bridge between the weak 2021 nIOD event and the strong 2022 nIOD. The pre-conditioning of TIO during 2021 by WWBs and La Niña was therefore important for the development of the 2022 nIOD event. The projected increase in the frequency of La Niña events (Geng et al., 2023; Wang et al., 2023) poses a challenge to the life, livelihood, and economy of India, as La Niñas tend to result in floods and damage to agriculture. The compensating effect of multi-year nIOD events can reduce the risks



associated with such La Niñas. Multi-year nIOD events can, however, suppress or enhance rainfall elsewhere, thereby translating the threat of compound extremes to other geographical locations.

**Code and data availability**

The data used in this study can be downloaded from the following websites:

1.  ERSST (https://psl.noaa.gov/data/gridded/data.noaa.ersst.v5.html)

2.  OISST (https://psl.noaa.gov/data/gridded/data.noaa.oisst.v2.html)

3.  HadISST (https://www.metoffice.gov.uk/hadobs/hadisst/)

4.  SOI
   (https://data.longpaddock.qld.gov.au/SeasonalClimateOutlook/SouthernOscillationIndex/SOIDataFiles/LatestSOI18
87-1989Base.txt)

5.  MJO phase and amplitude (https://iprc.soest.hawaii.edu/users/kazuyosh/ISO_index/data/MJO_25-90bpfil.rt_pc.txt)

6.  BSISO phase and amplitude (https://iprc.soest.hawaii.edu/users/kazuyosh/ISO_index/data/BSISO_25-90bpfil.rt_pc.txt)

7.  ERA5 reanalysis (https://www.ecmwf.int/en/forecasts/dataset/ecmwf-reanalysis-v5).

8.  NCEP GODAS (https://psl.noaa.gov/data/gridded/data.godas.html)

9.  MOAA GPV (https://www.jamstec.go.jp/argo_research/dataset/moaagpv/moaa_en.html)

10. SLA data from the Copernicus Marine Service website (http://climate.copernicus.eu)

The scripts used to produce the analyses and figures in this study are available on request from the authors.

**Author contributions**

Ankur Srivastava performed the analysis, figures, and wrote the initial draft of the manuscript. Suryachandra A. Rao conceived the original motivation for the study, and Gill M. Martin, Maheswar Pradhan, and Sarah Ineson contributed to the design of the study and the analyses. All authors contributed to the understanding and interpretation of the analyses, and to the final manuscript draft.

**Competing interests**

The authors have no competing interests to declare.



**Acknowledgements**

Gill M. Martin and Sarah Ineson were funded by the Met Office Weather and Climate Science for Service Partnership
(WCSSP) India project, which is supported by the UK Department for Science, Innovation and Technology (DSIT). WCSSP
India is a collaborative initiative between the Met Office and the Indian Ministry of Earth Sciences (MoES). Ankur
Srivastava, Maheswar Pradhan, and Suryachandra A. Rao are supported by the Indian Institute of Tropical Meteorology
(IITM), and IITM is fully funded by the Ministry of Earth Sciences, Government of India.

**Financial Support**

This work has been supported by the UK Department for Science, Innovation and Technology (DSIT) (Met Office Weather
and Climate Science for Service Partnership (WCSSP) India project).

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
