# Peer review of "The multi-year negative Indian Ocean Dipole of 2021-2022"

_EGUsphere, 2025_

## Author Comment (AC1)

*We would like to thank the reviewer for the assessment of our manuscript and for the recommendations for improvements. Responses to the comments and the proposed revisions to the manuscript are included below.*

**Main Comment #1**

In this paper, the authors emphasize the role of westerly wind anomalies, particularly those in January 2021 and the frequent, long-lived westerly wind bursts (WWBs) from 2021-2022, in sustaining the multi-year nIOD event of 2021-2022. However, based on Figure 2, it seems that other nIOD events also exhibit periods of strong or frequent westerly wind activity before or during the events.

a) Are the WWBs during 2021-2022 significantly stronger, longer-lasting, or more frequent than those in other nIOD events? A more quantitative comparison on winds between the 2021-2022 event and other nIOD events would help support the claim that the WWB occurrence in 2021-2022 is unusual.

***Response:***

To quantitatively assess the westerly anomalies associated with negative IOD events, we have counted the number of days in each month/3-month rolling aggregate when the westerly anomaly was greater than 2 m s$^{-1}$. The same is shown below as Figure R1. For the nIOD events of 1996, 2005, and 2010, the westerly anomalies are dominant towards the end of boreal summer (JAS, ASO). Some nIOD events develop early in the summer (such as 2016, 1998) and are associated with an early initiation of westerly anomalies in the equatorial Indian Ocean (EIO; Lu et al., 2016). Unlike all such events, the 2021-22 event has strong and persistent westerly anomalies in the equatorial Indian Ocean from OND 2020 onwards, and there are only 3 months between October 2020 and December 2022 where no such westerly anomalies occur. Also, Figure 3a quantifies that 2021 and 2022 had a record number and duration of WWBs. These two figures support our arguments about the role of persistent westerly anomalies and WWBs in sustaining the event.

Since, in our opinion, Figures 1, 2, and 3a highlight most of these aspects, we would like the reviewer's suggestion about whether to include one or the other panel of Figure R1 in the main manuscript.

[Figure]

Figure R1: The count of the number of days in (a) a month, and (b) 3-month rolling aggregate when the westerly wind anomaly averaged over the equatorial Indian Ocean (5°S-5°N, 60°E-90°E) was greater than 2 m s$^{-1}$. The wind anomalies for the previous two years are indicated by Year -2 and Year -1 on the x-axis.

b) How do we understand nIOD events that developed following strong easterly winds, as seen in late 1997, for example.

*Response:* The development of nIOD conditions is dependent on, amongst other factors, the timing and magnitude of the evolution of zonal westerly wind anomalies over the equatorial Indian Ocean. Prior to the development of nIOD conditions during 1998, there was an anomalous accumulation of subsurface oceanic heat content in the western tropical Indian Ocean due to the pIOD during 1997. A WWB event in the equatorial Indian Ocean during May 1998 caused the initiation of nIOD conditions during 1998 (Lim and Hendon, 2017). Westerly wind activity was pronounced from May to December for the 2016 nIOD event, which caused its early initiation during summer. Therefore, the timing and persistence of westerly wind anomalies in the equatorial Indian Ocean are important in determining the evolution of nIOD conditions. For the 2021-2022 case, the westerlies persisted during the preceding year as well (Figure 2). This preconditioned the Indian Ocean and sustained the nIOD event for a long duration.

c) This also relates to point (a). In line 130, the statement "each nIOD event has a unique evolution of zonal winds" suggests that wind patterns prior to nIOD events vary significantly. Does this imply that every nIOD event has a distinct formation mechanism? I think it would be helpful to clarify whether the 2021-2022 nIOD is a rare case due to specific conditions like long-lasting westerly wind bursts and Wyrtki Jets, or that nIOD events in general do not follow a common formation mechanism.

***Response:*** We wish to stress here that it is the timing and persistence of anomalous westerlies and WWBs, and their relation to ENSO conditions and evolution, that govern the evolution of nIOD conditions and can be unique for each event. For instance, as mentioned in response to (b) above, the 2016 event peaked during the boreal summer rather than fall due to early initiation of westerly anomalies in the equatorial Indian Ocean during May itself. Therefore, even though some events can have unique drivers (For e.g., the anomalous build-up of ocean heat content in the western Indian Ocean for the 1998 case mentioned in response to (b) above), the events do follow some common formation mechanisms in terms of the response to anomalous westerlies, WWBs, and Wyrtki jets. Therefore, we propose to rewrite lines 130-132 as:

"For most nIOD events, strong westerly wind anomalies are generally favored from mid-summer (July-August), persisting till the end of the year. However, the timing of the occurrence of anomalous westerlies in the TIO can impact the evolution of nIOD events. The 2016 event, for example, peaked during boreal summer due to the early initiation of anomalous westerlies (Lu et al., 2018). The 1998 nIOD was associated with a build-up of anomalous heat content in the western TIO, and a WWB event during May in the equatorial Indian Ocean caused the early initiation of nIOD. Also, the abrupt transition from the 1997 El Niño to a La Niña state during May 1998 (McPhaden, 1999) could have aided the nIOD event. Therefore, it is important to assess the evolution of zonal winds in TIO for the 2021-2022 event."

**Main Comment #2**

Lines 178–234 describe the proposed mechanisms and processes in detail, but some parts would benefit from clearer explanation or stronger supporting evidence.

For example, in lines 184-186, the statement that "this unusual occurrence of the WJ in January 2021 was due to the formation of a low-pressure system off the coast of Sri Lanka. This system led to widespread extreme rainfall over parts of Southern India" is interesting, but there is little evidence presented for the low-pressure system and its link to the WJ and rainfall.

***Response:*** The details of the extreme rainfall event over Peninsular India are discussed in detail in the severe weather bulletin issued by the India Meteorological Department (India Meteorological Department, 2021) and is cited at line 187. A cyclonic circulation formed over peninsular India and Sri Lanka during 11-15 January 2021 and moved westwards. The rainfall stations in the peninsular Indian state of Tamil Nadu recorded extremely heavy to heavy rainfall associated with this system. Under the influence of this cyclonic circulation, strong westerly wind anomalies set up to the south of this system, over the equatorial Indian Ocean. The low-level circulation and the OLR anomalies for the WWB event 1 are shown in Figure R2.

[Figure]

Figure R2: The OLR and 850 hPa wind anomalies for 13-01-2021.

We propose to rewrite lines 184-187 as follows:

"A cyclonic circulation formed over peninsular India and Sri Lanka during 11-15 January 2021 and moved westwards. The rainfall stations in the peninsular Indian state of Tamil Nadu recorded extremely heavy to heavy rainfall associated with this system (India Meteorological Department, 2021). Under the influence of this cyclonic circulation, strong westerly wind anomalies set up to the south of this system, over the equatorial Indian Ocean (figure not shown), and ultimately led to the unusual occurrence of the WWB on 13th January 2021."

Similarly, in lines 207-208, the claim that "the weakening of climatological south-easterlies along the Java-Sumatra coast during boreal summer reduced the evaporative cooling and maintained the positive SST anomaly" seems a bit speculative without enough supporting analysis. I would recommend either providing additional evidence or softening the language.

***Response:*** Figure R3 below shows that the latent heat flux anomaly over the IOD east pole region during JJA 2021 and 2022 was negative, along with the associated weakening of climatological south-easterlies along the Java-Sumatra coast. This is indicative of reduced evaporative cooling. Also, annual upwelling along the Java-Sumatra coast occurs from June to October (Susanto et al., 2001). The weakened south-easterlies over this region likely indicated reduced upwelling, which can maintain positive SST anomalies over the IOD east pole. The reduced upwelling is discussed in detail in Section 3.3. We propose rewording 207-208 as:

"The associated weakening of climatological south-easterlies along the Java-Sumatra coast during boreal summer reduced the evaporative cooling through the wind-evaporation-SST feedback (as indicated by latent heat flux anomaly, figure not shown), and suppressed upwelling (discussed in detail in Section 3.3) and maintained the positive SST anomaly. Thus, the negative IOD conditions were maintained during the boreal summer of 2021."

[Figure]

Figure R3: The anomalous JJA mean low-level circulation at 850 hPa and the latent heat flux anomalies for the years 2021 and 2022. A positive anomaly indicates reduced evaporative cooling.

**Minor Comments**

Lines 24-29:

a) This sentence seems to imply that negative IOD events correspond to drier conditions in Indonesia and Australia. However, these regions typically experience *increased* rainfall during negative IOD events. I suggest revising this part for clarity.

b) The impacts of IOD events on Sri Lanka, South China, and Brazil are not clearly described. It would be helpful to be more specific about the impacts on these regions.

**Response:** We propose rewriting this for clarity as below:

"Anomalously warm SST in the eastern Indian Ocean, along with cooler SSTs in the western Indian Ocean, characterize a negative Indian Ocean Dipole (nIOD) event. Such events are known to suppress the Indian Summer Monsoon Rainfall (ISMR; Ajayamohan et al., 2008, 2009; Ajayamohan & Rao, 2008; Ashok et al., 2001; Cherchi et al., 2007; Cherchi & Navarra, 2013; Krishnan et al., 2011; Ratna et al., 2021). nIOD events can also lead to severe droughts during the short rainy season in East Africa (Black et al., 2003; Endris et al., 2019; Manatsa & Behera, 2014), and enhance rainfall in Indonesia (Nur'utami & Hidayat, 2016) and Australia (Cai et al., 2009). pIOD events enhance the Maha rainfall in Sri Lanka (Zubair et al., 2003),

produce dipolar rainfall anomalies over Brazil, and enhance rainfall over South China (Bazo et al., 2013; Chan et al., 2008; Qiu et al., 2014; Taschetto & Ambrizzi, 2012)."

Lines 31-33:

a) Are the anomalies computed relative to a climatology? Please specify.

b) How is the standard deviation calculated for standardizing the DMI? Which baseline period is used for this?

c) Include references describing the method for computing the DMI.

***Response:***

The anomalies are computed with respect to the 1960-2022 climatology, and the standard deviation of the calculated index for 1960-2022 is used to standardize the index. The reference for calculating DMI is now included (Saji et al., 1999). We propose the following modification to the text:

"IOD conditions in the Indian Ocean are quantified using the dipole mode index (DMI; Saji et al., 1999), which is defined as the difference in area-averaged SST anomalies over western (50°E-70°E, 10°S-10°N) and eastern Indian Ocean (90°E-110°E, 10°S-Equator), normalized by its standard deviation. SST anomalies are calculated with respect to the 1960-2022 climatology in this study and are detrended. The indices hence calculated are normalized by the standard deviation of the time-series for 1960-2022."

Lines 37-39: Please specify what is meant by "never been associated with other nIOD events"

***Response:*** We propose rewriting this as below:

"Such persistent nIOD conditions, lasting for about 19 months during 2021-2022, have not been observed in association with any other nIOD event in the observational record since 1960 (since when reliable Indian Ocean observations became available, Fig. 1d)."

Line 58: 2021 was a normal monsoon year (99% of LPA)

***Response:*** This will be corrected as suggested: "2021 was a normal monsoon year (99% of LPA)."

Lines 60-61:

a) How about the modulation of the Walker Circulation?

b) Include references that discuss the role of the local Hadley Circulation and/or Walker Circulation.

**Response:**

(a & b) We propose adding the following discussions and references:

"IOD events are closely related to the ENSO state, with the relation between nIOD and La-Nina being much stronger compared to that between El-Nino and pIOD (Ashok et al., 2001; Saji and Yamagata, 2003). The lack of a large positive rainfall anomaly in these years could have been caused by the compensating effect of the nIOD conditions. Two competing processes might have co-existed: (i) The La-Nina associated enhancement of zonal Walker circulation, which tends to enhance rainfall over India by promoting convection over Indian longitudes, and (ii) anomalously warm SST in the IOD east pole can suppress the convection over India by the modulation of the local Hadley circulation. nIOD events are also associated with a strengthened Indian Ocean Walker Circulation, with enhanced convection over the eastern TIO (Lu et al., 2018)."

Lines 72-73: Clarify what specific "anomalous changes" are being referred to here.

**Response:** Proposed to be reworded to "anomalous changes to the volume transport of these jets"

Line 74: The question introduced here feels abrupt and disconnected from the following paragraph. It may not be necessary in my view, however, if you choose to keep it, consider adding how the previous studies have examined this question and/or if it will be addressed in this work.

**Response:** Since we do dwell upon the impact of La-Nina state on the nIOD, we propose to rewrite it and move it to the end of the paragraph:

"The 2021-22 nIOD event was, therefore, unique in several aspects: (1) It was the strongest nIOD event recorded to date. (2) It peaked during the boreal summer of 2022. (3) nIOD conditions were sustained for a record 19 months, making it the first occurrence of a multi-year nIOD event. (4) This event co-occurred with a triple-dip La Niña event. Did the triple-dip La Niña of 2020-2022 help in sustaining the nIOD conditions? This study will attempt to answer this question.

Figure 1:

a) I assume the different colors for NIOD, PIOD, and Neutral are based on standard deviation thresholds. Please clarify the criteria used.

b) What do the orange bars in Figure 1b represent?

***Response:***

(a) Yes, the IOD events selected here are based on whether the DMI index was more or less than 1 standard deviation. The caption for Figure 1 and the discussion at line 120 will be modified to reflect this.

"…….for all the nIOD events since 1980 (nIOD events are defined based on whether the normalized DMI was less than -1 standard deviation)."

"Figure 1: (a) The September-November (SON) mean DMI index, and the IOD east and west poles, for the period 1960-2022 based on ERSST dataset. IOD events are categorized as negative (positive) based on whether the normalized DMI was less than (greater than) -1 (+1) standard deviation. (b)………."

(b) The orange bars in Fig. 1b were inadvertent, will be removed.

Lines 75-77: Phrases like "the strongest recorded to date" and "the first occurrence of a multi-year nIOD" are quite strong. Please be specific, for example, clarify the time period (e.g., since when) these statements refer to.

***Response:*** We propose to reword this to:

"It was the strongest nIOD event recorded during 1960-2022." & "the first occurrence of a multi-year nIOD event during 1960-2022"

Line 127: zonal-mean?

***Response:*** Will be corrected to "Evolution of the equatorial 10m zonal mean wind anomalies".

Lines 158-161: This paragraph describes what is shown in Figures 4 and 5, and may be more appropriate for the figure captions. Consider moving the detailed descriptions (e.g., event numbers) to the captions and using the main text to focus more on interpretation or analysis.

***Response:*** Lines 158-161 will be moved to figure captions. Only the last line will be retained – "The succeeding discussion is arranged around the event numbers as shown in Figs. 4 and 5."

Line 169: The abbreviation "AVISO" is not defined before. Is this the same SLA product available from the Copernicus Marine Service website?

***Response:*** Yes, it is the same. Figure 4d caption will be reworded to "the observed Altimeter satellite gridded Sea Level Anomaly (SLA) level 4 data obtained from the Copernicus Marine Service website."

Line 173: The abbreviation "OSCAR" is not defined before.

***Response:*** Thanks for pointing these out. The currents shown in Figure 4 and 5 are from OSCAR (Ocean Surface Current Analysis Real-time) third degree dataset (DOI: 10.5067/OSCAR-03D01). Section 2, Data and Methods, and Figure 4 caption will be modified to reflect this:

"The ocean temperature profiles (pentad) are obtained from the National Centers for Environmental Prediction (NCEP) Global Ocean Data Assimilation System (GODAS; Behringer et al., 1998), and the depth of the 20 °C isotherm is computed from it. OSCAR (Ocean Surface Current Analysis Real-time, DOI:        10.5067/OSCAR-03D01) dataset provides near-surface ocean current estimates on a 1/3-degree grid with a 5-day resolution."

Figure 10: Include the units for both the vectors and shadings.

***Response:*** To be included in the figure caption as "The units are meters per second for the zonal component and Pascals per second for the vertical velocity."

**References:**

Ashok, K., Guan, Z., and Yamagata, T.: Impact of the Indian Ocean Dipole on the Relationship between the Indian Monsoon Rainfall and ENSO, Geophys Res Lett, 28, 4499–4502, https://doi.org/10.1029/2001GL013294, 2001.

India Meteorological Department: Brief Report on Extremely Heavy Rainfall Events over Tamil Nadu (1-15 Jan 2021), 1–32 pp., 2021.

Lim, E. P. and Hendon, H. H.: Causes and Predictability of the Negative Indian Ocean Dipole and Its Impact on la Niña during 2016, Sci Rep, 7, https://doi.org/10.1038/s41598-017-12674-z, 2017.

Lu, B., Ren, H. L., Scaife, A. A., Wu, J., Dunstone, N., Smith, D., Wan, J., Eade, R., MacLachlan, C., and Gordon, M.: An extreme negative Indian Ocean Dipole event in 2016: dynamics and predictability, Clim Dyn, 51, https://doi.org/10.1007/s00382-017-3908-2, 2018.

McPhaden, M. J.: Genesis and evolution of the 1997-98 El Nino, https://doi.org/10.1126/science.283.5404.950, 1999.

Saji, N. H. and Yamagata, T.: Possible impacts of Indian Ocean Dipole mode events on global climate, Clim Res, 25, 151–169, https://doi.org/10.3354/cr025151, 2003.

Saji, N. H., Goswami, B. N., Vinayachandran, P. N., and Yamagata, T.: A dipole mode in the tropical Indian Ocean, Nature, 401, 360–363, https://doi.org/10.1038/43854, 1999.

Susanto, R. D., Gordon, A. L., and Zheng, Q.: Upwelling along the coasts of Java and Sumatra and its relation to ENSO, Geophys Res Lett, 28, https://doi.org/10.1029/2000GL011844, 2001.

---

## Author Comment (AC2)

*We would like to thank the reviewer for the assessment of our manuscript and for the recommendations for improvements. Responses to the comments and the proposed revisions to the manuscript are included below.*

The manuscript is solid study of the different contributions to the multi-year IOD event of 2021-2022 and a good timeline of how the 2021-22 event came to be. However, the manuscript's stated goal is to explore how unusual the characteristics of the 2021-22 IOD event are. Here, the manuscript fails to explore in greater detail the uniqueness of the 2012-22 period or its similarities to other multiyear IOD events in the record. Part of this is the record length. For the detailed part of the study, many of the datasets used are not long enough to get produce a detailed exploration of the extended IOD event and its relationship with the unusual multi-year La Niña event in the Pacific. However, even without all the detail, some of the reanalysis products used do have extended records that would allow for comparisons of other extended IOD events and the potential for co-occurrence of extended La Niña events. In addition to looking at canonical definitions of La Niña, I would recommend considering using the relative Niño Index (van Oldenborgh et al 2021), for a better understanding of ENSO state and climate change and La Niña in relationship to the rest of the tropics. Further, an expanded exploration of long lasting La Niña and IOD events would also provide more information on the importance of multi-year events for Indian Summer Monsoon Rainfall and how anomalous summer rainfall was during this event compared with other extended IOD or ENSO events as discussed in section 3.4.

Sources:

Van Oldenborgh GJ, Hendon H, Stockdale T, L'Heureux M, De Perez EC, Singh R, Van Aalst M. Defining El Niño indices in a warming climate. Environmental Research Letters. 2021 Mar 11;16(4):044003

**Response:** We read "2012-22 period or its similarities....." the reviewer's comment as "2021-22 period or its similarities....." as we believe 2012 is a typo.

1. **Record length and uniqueness of 2021-22 event**:
    a. We focus on the occurrences of nIOD events in the 1960-2022 period, as systemic observations of the Indian Ocean started later compared to other ocean basins (Masumoto et al. 2010; Zeng et al., 2020). Indian Ocean exploration started in the 1960s with the first International Indian Ocean Expedition (Knauss, 1961) and was followed by various other exploration programs. Extended records include periods with sparse observation, and even the tropical Pacific has sparse observations prior to about 1955 (D'Arrigo et al., 2008; Kaplan et al., 1998). Thus, studies of the Indian Ocean often utilize data post-1960s (for eg. Alory et al., 2007; Murtugudde et al., 2000; Nyadjro et al., 2013; Saji et al., 1999; Zhang and Du, 2021).

**b.** However, following the reviewer's suggestion, we verified our findings in a longer observational record. ERSST data for the period 1900-2022 is used to compute the standardized DMI index from detrended SON SST anomalies. We categorize the events as nIOD if the standardized DMI is less than -1. Using these nIOD events, we count the length of contiguous periods (in months) with monthly DMI < -0.5 (shown in Fig. R1). For example, if a particular year is categorized as an nIOD event based on SON DMI, we calculate the duration of that event based on the number of contiguous months where the index remained less than -0.5. It is evident that 2021-2022 was the only event during 1900-2022 when the DMI < -0.5 for more than 12 months at a stretch.

In view of (a) and (b) above, we propose to retain the period of analysis as 1960-2022, as in the original manuscript.

[Figure]

**Fig. R1** The length of contiguous periods (in months) with monthly DMI < -0.5 for the nIOD events in the period 1900-2022. The 2021-22 event is highlighted in red.

There are no other such prolonged nIOD events to compare with in the 1960-2022 period, or even in a longer 1900-2022 period. The other unique features of this event are highlighted in the manuscript in terms of unusual occurrences of anomalous westerlies over the equatorial Indian Ocean (Fig. 2), record occurrences of WWBs (Fig. 3), anomalous sub-surface ocean conditions (Figs. 7 & 8), and the associated discussions.

2.  **Use of relative Niño Index** – Thank you for the suggestion. We propose to modify Fig. 1c to include the monthly relative oceanic Niño index (RONI) obtained from https://www.cpc.ncep.noaa.gov/data/indices/. Modified Fig. 1 (b & c) is reproduced below. Using the RONI index does not change our inferences.

[Figure]

**Fig. R2** The monthly DMI index from ERSST (bars), and the three-month running mean for ERSST, OISST, and HadISST (lines) for the period 2020-2022 (top-panel); bottom-panel same as top-panel but for the Nino 3.4 index. Also shown in the bottom panel is the monthly relative oceanic Niño index (RONI) obtained from https://www.cpc.ncep.noaa.gov/data/indices/.

We propose to rewrite the discussion at line 41 to reflect the use of the RONI index, the associated reference, and in Section 2, Data and Methods.
Line 40 "(as is seen from the Niño 3.4 index, and the relative Niño 3.4 index, Fig. 1c)."
Line 100 "The relative Nino 3.4 index (L'Heureux et al., 2024; Van Oldenborgh et al., 2021) is obtained from https://www.cpc.ncep.noaa.gov/data/indices/"

3.  Exploration of other long-lasting La Niña and IOD events – As seen from Fig. R1 above, there were no other prolonged nIOD events, even for an extended period of 1900-2022. Therefore, it is not possible to compare the 2021-22 nIOD event with other events. The impact of long-lasting La Niña events has been explored in recent studies (Ratna et al., 2024; Sharma et al., 2024). We propose to add these references to the manuscript at lines 56-57.

"La Niña years are associated with notably decreased (increased) ISMR when preceded by La Niña (El-Nino) conditions during the previous winter (Sharma et al., 2024), and triple-dip La Niña events are often associated with positive ISMR anomaly (Ratna et al., 2024)"

**References**

Alory, G., Wijffels, S., and Meyers, G.: Observed temperature trends in the Indian Ocean over 1960–1999 and associated mechanisms, Geophys Res Lett, 34, https://doi.org/10.1029/2006GL028044, 2007.

D'Arrigo, R., Allan, R., Wilson, R., Palmer, J., Sakulich, J., Smerdon, J. E., Bijaksana, S., and Ngkoimani, L. O.: Pacific and Indian Ocean climate signals in a tree-ring record of Java monsoon drought, International Journal of Climatology, 28, 1889–1901, https://doi.org/10.1002/JOC.1679;PAGEGROUP:STRING:PUBLICATION, 2008.

Kaplan, A., Cane, M. A., Kushnir, Y., Clement, A. C., Blumenthal, M. B., and Rajagopalan, B.: Analyses of global sea surface temperature 1856-1991, J Geophys Res Oceans, 103, https://doi.org/10.1029/97JC01736, 1998.

Knauss, J. A.: The International Indian Ocean Expedition, Science (1979), 134, https://doi.org/10.1126/science.134.3491.1674, 1961.

L'Heureux, M. L., Tippett, M. K., Wheeler, M. C., Nguyen, H., Narsey, S., Johnson, N., Hu, Z. Z., Watkins, A. B., Lucas, C., Ganter, C., Becker, E., Wang, W., and Di Liberto, T.: A Relative Sea Surface Temperature Index for Classifying ENSO Events in a Changing Climate, J Clim, 37, https://doi.org/10.1175/JCLI-D-23-0406.1, 2024.

Murtugudde, R., McCreary, J. P., and Busalacchi, A. J.: Oceanic processes associated with anomalous events in the Indian Ocean with relevance to 1997-1998, https://doi.org/10.1029/1999jc900294, 2000.

Nyadjro, E. S., Subrahmanyam, B., and Giese, B. S.: Variability of salt flux in the Indian Ocean during 1960–2008, Remote Sens Environ, 134, 175–193, https://doi.org/10.1016/j.rse.2013.03.005, 2013.

Van Oldenborgh, G. J., Hendon, H., Stockdale, T., L'Heureux, M., Coughlan De Perez, E., Singh, R., and Van Aalst, M.: Defining El Nio indices in a warming climate, Environmental Research Letters, 16, https://doi.org/10.1088/1748-9326/abe9ed, 2021.

Ratna, S., Musale, M., Sharma, T., C. T., S., Rohini, P., Bandgar, A., Sreejith, O. P., and Hosalikar, K. S.: Triple-dip La Niña (2020-2022) and its impact on Indian Summer Monsoon Rainfall: Insight from the Monsoon Mission Coupled Forecasting System, MAUSAM, 75, 759–768, https://doi.org/10.54302/mausam.v75i3.6283, 2024.

Saji, N. H., Goswami, B. N., Vinayachandran, P. N., and Yamagata, T.: A dipole mode in the tropical Indian Ocean, Nature, 401, 360–363, https://doi.org/10.1038/43854, 1999.

Sharma, T., Ratna, S. B., Pai, D. S., Bandgar, A., Rajeevan, M., Mohapatra, M., Sreejith, O. P., and Hosalikar, K. S.: Indian summer monsoon rainfall response to two distinct

evolutions of La Niña events, International Journal of Climatology, 44, 4405–4427, https://doi.org/10.1002/joc.8588, 2024.

Zeng, L., Chen, G., Huang, K., Chen, J., He, Y., Zhou, F., Yang, Y., Liang, Z., Peng, Q., Shi, R., Gamage, T. P., Chen, R., Li, J., Zhang, Z., Wu, Z., Yu, L., and Wang, D.: A decade of eastern tropical Indian Ocean Observation Network (TIOON), Bull Am Meteorol Soc, 101, https://doi.org/10.1175/BAMS-D-19-0234.1, 2020.

Zhang, Y. and Du, Y.: Extreme IOD induced tropical Indian Ocean warming in 2020, https://doi.org/10.1186/s40562-021-00207-6, 2021.

---

## Author Response (AR2)

*We would like to thank the reviewer and the co-editor for their assessment of our manuscript and for their recommendations for improvements. Responses to the comments and the proposed revisions to the manuscript are included below, along with the Line numbers at which the suggestions are incorporated. All line numbers refer to the tracked-changes version of the manuscript.*

The author has addressed my previous comments, especially the additional explanation in Section 3.1 of how 2021-2022 differs from previous nIOD events is particularly helpful and highlights the uniqueness of the 2021–2022 nIOD event.

I still have one suggestion regarding the relationship between WWBs and La Niña forcing (details below), along with a few minor comments/edits for your reference. All line numbers below refer to the tracked-changes version of the manuscript.

--

In the revision (L243–248), you suggest that the unusual WWB in January 2021 may be related to cyclonic circulation over peninsular India and Sri Lanka. This is an interesting point that may be worth mentioning in the conclusion section.

*Response: Now included at Line 387.*

I also feel that the statement, "Out of the eight WWB events during 2021–22, five occurred when SOI was greater than +10, suggesting the role of La Niña forcing (conclusion #5)," is a bit too strong, as five out of eight does not constitute strong evidence. In my view, within this study, the drivers of WWBs remain somewhat ambiguous (as you note, cyclonic circulation may also have contributed to the unusual WWB). Accordingly, the extent to which La Niña forcing "plays an important role" in maintaining WWBs, relative to other factors, remains uncertain, and you may wish to soften some of these statements.

*Response: Thank you for the suggestion. We have incorporated the following changes to reflect this.*

*Line 12: "westerlies were **possibly** supported by the background...*

*Lines 399-403: "WWBs played an important role in sustaining the multi-year nIOD event. La Niña forcing played an important role in maintaining conducive conditions for WWBs. Out of the eight WWB events during 2021-22, five occurred when SOI was greater than +10, suggesting the **possible** role of La Niña forcing. Some of the peaks in SOI match well with the peaks of anomalous westerly wind activity, while in other instances, the WWB activity occurred during the strengthening/mature phase of the SOI. **A more quantitative assessment of the impact of La Niña on WWBs requires further investigation**."*

*Lines 414-415: "The La Niña modulated Walker circulation, **thus providing favourable conditions** for WWBs in the TIO."*

--

L28: "pIOD" has not been defined.

*Response: Defined now at Line 25 of the revised manuscript.*

L248: Why specify the WWB on 13th January 2021 when a WWB is at least a 4-day event (as defined in Section 2), if not longer?

*Response: Thanks for noticing this. This is now reworded to "led to the unusual occurrence of the WWB centred around 13$^{th}$ January 2021"*

L334: It was the number and duration that were unusual compared to other years – would it help to refer to Figure 2c here?

*Response: Now referred to at line 301 of the revised manuscript.*

L344: Out of the eight WWBs during 2021-22, six out of eight events were ... – delete the second out of eight to avoid duplication.

*Response: Deleted at line 311.*

Figure 5. I can't tell the BSISO phase in event #7.

*Response: The BSISO phase is 2; Fig. 5 is now slightly modified to show it clearly.*

L379-380: Please explain why "The large sub-surface heat content during DJF and MAM 2021 was therefore not reflected in the SST."

*Response: Now provided at lines 347 – "The large sub-surface heat content during DJF and MAM 2021 was therefore not reflected in the SST due to the negative coupling between SST and D20 (Fig. 9)."*